# Polycomb Assemblies Multitask to Regulate Transcription

**Miguel Vidal**

Department of Cellular and Molecular Biology, Centro de Investigaciones Biológicas, Ramiro de Maeztu 9, 28040 Madrid, Spain; mvidal@cib.csic.es

**Abstract:** The Polycomb system is made of an evolutionary ancient group of proteins, present throughout plants and animals. Known initially from developmental studies with the fly *Drosophila melanogaster*, they were associated with stable sustainment of gene repression and maintenance of cell identity. Acting as multiprotein assemblies with an ability to modify chromatin, through chemical additions to histones and organization of topological domains, they have been involved subsequently in control of developmental transitions and in cell homeostasis. Recent work has unveiled an association of Polycomb components with transcriptionally active loci and the promotion of gene expression, in clear contrast with conventional recognition as repressors. Focusing on mammalian models, I review here advances concerning roles in transcriptional control. Among new findings highlighted is the regulation of their catalytic properties, recruiting to targets, and activities in chromatin organization and compartmentalization. The need for a more integrated approach to the study of the Polycomb system, given its fundamental complexity and its adaptation to cell context, is discussed.

**Keywords:** Polycomb; chromatin modifier; PRC1; PRC2; transcriptional repression; histone E3 ligase; histone lysine methyltransferase; chromatin topology; nuclear condensates; DNA binding

---

## 1. Introduction

Specific and coherent gene expression patterns are essential to cell diversity of multicellular organisms. Such a selective use of the encoding potential of genomes is achieved largely through transcriptional regulation. Much of this function relies on transcription factors, proteins that bind DNA with both low and high affinities. Eukaryotic genomes are organised as chromatin, a lineal succession of nucleosomal units resulting from wrapping DNA around histones, thus precluding binding sites from access to transcription factors. This arrangement, further elaborated as high-order and topological domains poses important regulatory opportunities through the modulation of chromatin structure. This is accomplished by the action of chromatin regulators, a vast collection of multiprotein complexes endowed with catalytic and other activities, affecting nucleosomal dynamic or their chemical modification.

Within the ample collection of chromatin regulators present in eukaryotes, the Polycomb system has achieved a major prominence due to the nature of the genes under its regulatory influence, many of them important developmental regulators. Precisely, the morphological, homeotic alterations associated to the derepression of *Hox* genes, observed during the genetic analysis of the development of the fly *Drosophila melanogaster*, constitute the foundation of the discovery of the Polycomb group (PcG) of genes. Both circumstances, their activity in *D. melanogaster* and its initial association with gene repression has come to impact enormously how the study of the Polycomb system is approached. Its subsequent association with the repression of tumor suppressors and oncogenic transformation in mammals prompted further the interest in the system. It may appear paradoxical, however, that while

these early functions link the Polycomb system with transcriptionally silent states, a large number of genomic sites occupied by Polycomb are transcriptionally active, and that in certain cell types even the expression of some of them depends on the presence of Polycomb products.

Organised in biochemically heterogeneous, evolutionarily conserved protein complexes, the Polycomb machinery carries out a diversity of functions in transcriptional control and genome stability. Here I review recent advances in its architecture and its recruiting to targets regarding transcriptional functions, with an emphasis in metazoans and mammals in particular. Comprehensive reviews of plant PcGs can be found in [1,2]. Although the activities of the Polycomb system are best understood in the light of the products of the Trithorax group (TrxG) of genes, also identified during the genetic analysis of *Drosophila* development as antagonists of Polycomb mutations, the TrxG system will not be covered in any detail. Excellent TrxG-centered reviews can be found in [2–5].

## 2. Polycomb Complexes

The products of the PcG of genes form multiprotein assemblies termed after their best known transcriptional activity as Polycomb Repressive Complexes (PRCs). From plants to metazoans, PRCs are a collection of biochemical entities containing either of two catalytic modules endowed with histone modifying activity. Not all subunits present in the distinct PRCs correspond to products of (genetically identified) PcG genes. Conversely, some of the PcG gene products are not subunits of Polycomb complexes. An example of the latter is *O*-GlcNAc transferase (OGT), the product of *Drosophila Super sex combs*, an enzyme that adds *N*-acetylglucosamine to a multitude of proteins, some of which are PRC subunits [6]. PRCs are made of an obligate core complex, the catalytic module, and sets of stably associated subunits. Often, the complexes contain substoichiometric components and loosely bound proteins normally not considered Polycomb subunits.

Depending on the nucleosomal substrate they modify, two major classes of PRCs are recognised: PRC1, that contains an E3 ubiquitin ligase that monoubiquitylates lysine 119 of histone H2A (H2AK119Ub) [7], and PRC2 (found later) that contains a lysine methyl transferase (KMT) responsible for all methylated forms of lysine 27 in histone H3 tail (H3K27me1,2,3) [8–11]. The catalytic activities of these modules are subject to complex regulation but it is important to note that some Polycomb functions are independent of their catalytic activities (see below). Some PcG gene products form part of complexes that have diverged functionally between flies and mammals and are not considered here. It is the case of the fly deubiquitinase Calypso (homolog of vertebrate BRCA1 Associated Protein-1 (BAP1) in the Polycomb repressive deubiquitanse (PR-DUB) complex [12,13]. Thus, it is required for repression of *Hox* genes in flies while it protects genes from PRC1 repression in mammals [12,14]. Moreover, although it intersects some Polycomb functions, it removes ubiquitin not only from histone H2A but from many other non-PcG proteins [15].

PRC1 and PRC2 can be thought of as catalytic modules made of a core of subunits shared within either class of complexes, to which a variety of subunits associate, frequently in a mutually exclusive fashion, thus making the heterogenous collection of assemblies that PRCs are. Cell context-dependent combinations of subunits results in sets of related but different PRCs. Recent surveys of PcG subunits, functional domains and evolutive relationship can be found in [4,16]. Here, the essentials on composition and structure of stable PRCs is summarized, based mostly on mammalian sources. A simplified, consensual version of PRC1 and PRC2 complexes is depicted in Figure 1, and their subunits and relevant domains are listed in Table 1.

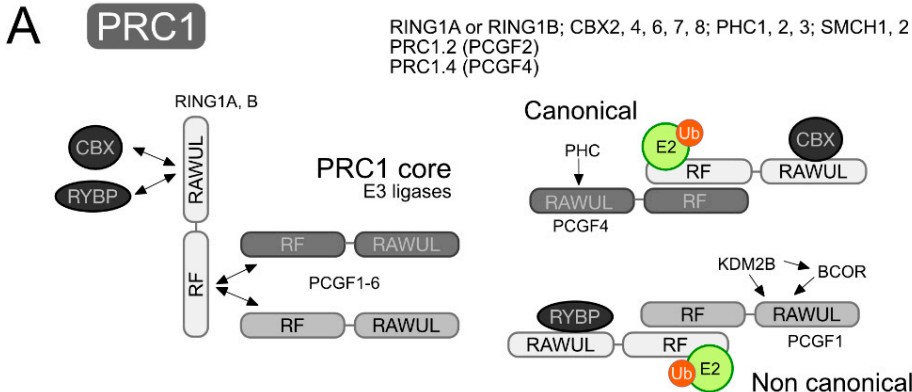

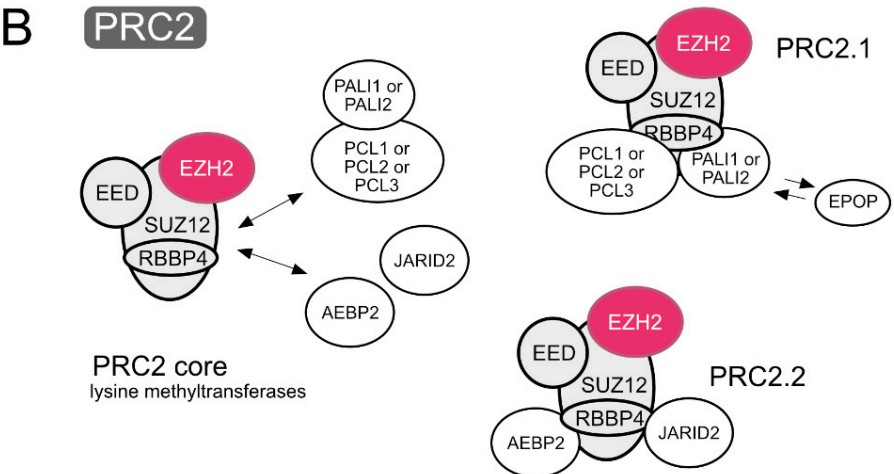

**Figure 1.** Schematic representation of Polycomb complexes, showing core components and accompanying subunits. (**A**) PRC1 cores are are heterodimeric RING-type E3 ligases made of a RING1 protein, RING1A or RING1B, and a PCGF subunit (PCGF1–6). While they contact each other through their RING finger motifs (RF), their C-terminal RING finger- and WD40-associated Ubiquitin-like (RAWUL) domains mediate selective associations with other PRC1 subunits. Which of the PCGFs is present in the heterodimer determines the subunits bound, thus defining six major types of complexes, PRC1.1 to PRC1.6. The RAWUL domain of RING1 proteins binds in exclusive fashion either chromobox homolog proteins (CBX) or Ring1 and YY1 Binding Protein (RYBP)/YY1-Associated Factor 2 (YAF2) subunits, characteristic of canonical and non-canonical PRC1 complexes, respectively. Activated ubiquitin (Ub) is depicted onto an E2 ligase contacting the RING1 component of the E3 ligase module that acts as an adaptor providing the substrate in the enzymatic modification of histone H2A. (**B**) The PRC2 core is made of Retinoblastoma Binding Protein 4 (RBBP4, or almost identical RBBP7), Embryonic Ectoderm Development (EED) and Suppressor of zeste 12 (SUZ12), obligate partners of the catalytic SET-lysine methyl transferase (KMTase) subunit Enhancer of zeste (EZH1 or EZH2). Two major classes of PRC2 complexes are defined by the mutually exclusive association of DNA-binding proteins: PRC2.1 complexes that contain a member of the Polycomb-like (PCL) family and PRC2.2 complexes that contain Adipocyte Enhancer-Binding Protein 2 (AEBP2) and Jumonji, AT rich interactive domain 2 (JARID2) subunits. An additional subunit, Elongin BC and polycomb repressive complex 2 associated protein (EPOP), can be found in PRC2.1 complexes, instead of PRC2 associated LCOR isoforms 1-3 (PALI1-3). The mutually exclusive nature of some associations is indicated by double arrows.

**Table 1.** Polycomb group (PcG) proteins, complexes, domains and major functions as chromatin regulators.

| Subunits | | | Domains | Chromatin Function |
|---|---|---|---|---|
| Mammals | Also Named | Flies | | |
| **Core PRC1—Common to All PRC1 Complexes** | | | | |
| RING1A, RING1B | RING1(A), RNF2(B) | dRing1/Sce | RING finger, RAWUL | E3 ubiquitin ligase |
| PCGF1-PCGF6 | BMI1(PCGF4), MEL18 (PCGF2) | | RING finger, RAWUL | E3 ubiquitin ligase |
| **Canonical PRC1 Complexes** | | | | |
| PHC1–PHC3 | | Ph | SAM | Oligomerization |
| CBX2, CBX4, CBX6–CBX8 | M33(CBX2), PC2(CBX4) | Pc | chromodomain | H3K27me3 binding |
| SMCH1, SMCH2 | | Scm | SAM, MBT | Oligomerization |
| SFMBT1, SFMBT2 | | Sfmbt | SAM, MBT | Oligomerization |
| L3MBTl1-L3MBTL4 | | L(3)mbt | SAM, MBT | Oligomerization |
| **Non-Canonical PRC1 Complexes** | | | | |
| RYBP/YAF2 | | dRybp | Zinc finger | Ub binding |
| KDM2B | FBXL10 | Kdm2 | CXXC, JmJC | DNA binding, H3K36 demethylase |
| MGA–MAX | | (1) | DNA binding motif | DNA binding |
| E2F6–TFDP1 | | (1) | DNA binding motif | DNA binding |
| WDR5 | | (1) | WD40 | scaffold? * |
| DCAF7 | WDR68 | (1) | WD40 | scaffold? * |
| **Core PRC2 Complex** | | | | |
| EZH1, EZH2 | | E(z) | SET | Lysine methyltransferase |
| SUZ12 | | Suz12 | VEFS box, Zinc finger | Allosteric integration |
| EED | | Esc, Escl | WD40 | H3K27me3 binding |
| RBBP4, RBBP7 | | Nurf55 | WD40 | H3K4me, H3K36 binding |
| **PRC2 Accessory Subunits** | | | | |
| PCL1–PCL3 | PHF1, MTF2, PHF19 | Pcl | Winged helix, Tudor | DNA binding, H3K36me binding |
| JARID2 | | Jarid2 | DNA binding motif, Ub binding motif | DNA binding, nucleosome binding |
| AEBP2 | | Jing | | DNA binding |
| EPOP | C17orf96 | (1) | | Low level gene expression |
| PALI1, PALI2 | C100rf12 | (1) | | Link to corepressors? * |

(1) No PcG-related homolog identified. * Likely function, not fully confirmed.

*2.1. PRC1 or Type I of Polycomb Repressive Complexes*

Initial PRC1 preparations, containing the products of characteristic *Drosophila* PcG genes or their mammalian homologs, are known as canonical complexes, to distinguish them from a distinct, new set of (non-canonical) PRC1 complexes identified later on. All PRC1 assemblies share a heterodimeric

E3 ubiquitin ligase, made of combinations of two types of RING finger-containing paralogs. One of them is made of RING1/RING1A and RNF2/RING1B, homologs of *Drosophila Sex combs* extra (*Sce*). The other, by either of the six products of the family of the Polycomb group of genes (PCGFs), whose RING fingers appear conserved in *Drosophila Posterior sex combs* (*Psc*), *Suppressor of zeste 2* (*Su(z)2*) or *Lethal* (*3*) *73Ah* ((*l*)(*3*)*73Ah*). The RING finger structure is characterised by a C3HC4 motif that binds two zinc cations in a way that builds a globular, compact domain apt for protein–protein interactions [17]. RING1 and PCGF subunits are structurally related proteins, with N-terminal RING fingers separated by unique, non-structured sequences from a C-terminal module folded as a ubiquitin like structure (Ring-finger and WD40 associated Ubiquitin-Like (RAWUL) domain [18]). Both well-defined folded regions participate in essential contacts with other Polycomb subunits. These heterodimers are held together mostly through extensive interactions through their RING fingers [19,20]. Given the rather selective association of non-RING finger PRC1 subunits with every heterodimeric E3, the presence of a given Polycomb Group RING Finger (PCGF) family member (PCGF1 to PCGF6) is used to name sets of PRC1 complexes as PRC1.1 to PRC1.6 [21]. Bound to the C-terminal ubiquitin (Ub)-like domains of RING1 subunits, all complexes contain RYBP, or its paralog YAF2 [21,22]. This minimal architecture is considered the core element of PRC1 complexes, and is present in all of the assemblies of the non-canonical class. Canonical PRC1, instead, lack RYBP or YAF2, which are replaced by a member of a subset of the chromobox homolog protein family (CBX2/M33, CBX4/PC4, CBX6, CBX7 and CBX8). This is due to the mutually exclusive nature of the interaction of RYBP–YAF2 or CBX proteins with the Ub-like domain of RING1 proteins [23]. CBX subunits are the products homologs of *Drosophila* Polycomb (PC). For reasons that are not clear, CBX-containing PRC1 complexes usually contain either PCGF2/MEL18 or PCGF4/BMI1 but no other PCGF members. It is important to note that the PRC1 architecture described above, as that of PRC2 (2.2), is a version distilled from evidence obtained from a rather reduced number of cell types. As research expands into other cell models or developmental times, variant PRC1 complexes may appear that do not conform fully to the above picture. Examples are the presence of CBX proteins associated to subunits of non-canonical complexes [24,25].

The identity of the PCGF member in the heterodimer decisively influences the nature of other subunits that associate with the core complex, in agreement with a modular organisation of PRC1 deduced from reconstitution experiments (Figure 1A). For example, assembling recombinant subunits of PRC1.1 in the presence/absence of BCL-6 corepressor (BCOR) shows the stable formation of a PCGF1–RING1B–RYBP core [26,27] and that the interface resulting from the association of PCGF1 and BCOR determines the association of lysine demethylase (KDM2B)/FBXL10 [27]. Likewise, a short segment in the Ub-like domain of PCGF4–BMI1 determines binding of Polyhomeotic-like protein 2 (PHC2) [28], one of the paralogs of the mammalian family with homology to *Drosophila* Polyhomeotic (PH). The relevant BMI1 sequence is conserved in the Ub-like domain of PCGF2–MEL18, the other PCGF family member of canonical PRC1 that also binds PHC2 [28]. It is not clear why RYBP-containing PRC1 complexes with PCGF2 or PCGF4 lack PHC subunits. Often, the combination of PCGF2/4, together with CBX and PHC paralogs, is taken as the defining signature of canonical PRC1 complexes. Unlike PCGF1, 4 and 6, no critical surfaces have been identified yet in PCGF homologs present in other PRC1 complexes. For example, in PRC1.6, PCGF6 seems to act as a bridge between the heterodimeric DNA-binding component MGA–MAX, and other subunits of the complex [29–31]. In general, PRC1 complexes locate on the genomes in non-overlapping patterns [32–34], pointing at a functional diversity still poorly characterized.

## 2.2. PRC2 or Type 2 of Polycomb Repressive Complexes

The catalytic core of PRC2 is rather homogeneous compared to the variety of PRC1 E3 modules resulting from iterations of RING and PCGF pairs. In addition to the lysine methyl transferase (KMT) (EZH2, EZH1), the enzymatic module of PRC2 contains subunits EED, SUZ12 and one of the extremely similar paralogs RBBP4, RBBP7. In cells expressing both EZH paralogs, PRC2 complexes contain one or the other [35,36]. EZHs and SUZ12 are large, multi-domain proteins displaying a multiplicity of

contacting surfaces. EED and RBBP4/7, on the other hand, are WD40 repeat proteins that fold in the characteristic propeller-shaped structure found in a multitude of WD40 proteins acting as adaptors or scaffolds mediating protein–protein interactions [37]. The activity of this multisubunit enzyme is modulated by the presence of additional subunits, many of which associate with the core in a mutually exclusive manner (Figure 1B). Depending on the content in these "accessory subunits" two major categories of complexes are distinguished, PRC2.1 and PRC2.2. The presence of any member of the family of Polycomb-like paralogs (PCL1-3) identifies PRC2.1 complexes. Additionally, either PALI1 (PRC2 associated ligand-dependentcorepressor (LCOR) isoform 1, one of the transcripts of the *LCOR* gene, which also encodes C10ORF12, a shorter form of PALI1) or PALI2 (encoded by the *LCORL* gene) [38] can be found in PRC2.1. Variant PRC2.1 complexes can contain EPOP instead of PALI1 [13,38,39]. PRC2.2 complexes, on the other hand, contain AEBP2 and JARID2, whose presence precludes that of PCL paralogs or PALI1. All these interactions do not involve well defined domains but, on the contrary, multiple contacts [29,38] and occupancy of surfaces by one of the subunits hinders the stable association of the competing subunit. Similar to PRC1 complexes, the functional capabilities of distinct PRC2 complexes may differ, not only by the presence of one or another type of DNA-binding domain (see below) but also by protein–protein interactions dictated by the "accessory subunits". For instance, PALI1 could intermediate interactions with the histone-KMT Euchromatic histone-lysine N-methyltransferase 2 (EHMT2)/G9a [38] whereas EPOP confers an ability to bind components of the transcription elongation factor Elongin BC complex [40,41]. Coexisting PRC2.1 and PRC2.2, at least in embryonic stem cells (ESCs), appear to act somehow antagonistically [38] and their activities are expected to vary within distinct developmental and differentiation settings.

*2.3. Dynamic, Cell-Context Dependent Display of Polycomb Complexes*

It is important to notice that the descriptions of PRC1 and PRC2 complexes correspond to the forms found in abundant cell types such as cultured cell lines: human (Human embryo kidney (HEK)) 293 [8,11,13,21,42], HeLa [43,44] and K562 [34]; murine ESCs [25,29,38,45–47] and hematopoietic types such as MEL [48] and 32D [49] or *Drosophila* embryos [9,10,50]. In vivo, the display of PRC complexes, the composition of their enzymatic modules and associated subunits and the relative proportions of each of the assemblies are cell-dependent. The steady state level of PRC subunits is regulated by both transcriptional and post-transcriptional means, sometimes involving cross-regulatory interactions between Polycomb products. For example, RING1B represses genes encoding PRC1 subunits CBX2, CBX4, CBX8, PHC2, and BMI1 in pluripotent ESCs, where accordingly they accumulate at very low levels [51], thus making PRC1.6 subunits the predominant partners of RING1B in these cells [29]. In cancer cells, PRC2 sustains levels of PRC1 complexes by preventing the upregulation of microRNAs (miRNAs) that downregulate PRC1 subunits [52]. When ESCs differentiate (to embryo bodies or neural or mesodermal progenitors) the levels of PRC1 subunits change and new complexes are present [29,33,51]. A singularly well documented case is that of CBX7, the prevalent CBX subunit in canonical PRC1 complexes in undifferentiated ESCs, which is downregulated by the accumulation of members of the *miR-125* and *miR-181* families of miRNAs [53]). A similar switch is also observed during the differentiation of early hematopoietic progenitors into multipotent progeny of lower self-renewal capacity [54]. For PRC2 complexes, a significant compositional alteration is the inversion in the relative levels of EZH paralogs that accompany differentiation in some cell lineages (hematopoietic, myogenic) by which EZH1 levels increase at the expense of those of EZH2 [55–57]. Substitution of EZH2 by EZH1 in PRC2 complexes is functionally relevant, considering the poor ability of EZH1 complexes to catalyse the deposition and spreading of the H3K27me3 modification [58,59].

Quantitative proteomics analysis during the generation of neural progenitors from ESCs, a widely used model of in vitro cell differentiation, illustrates accurately changes in PRC composition associated to cell identity [29]. Thus, while the relative proportions of core PRC2 subunits vary little with differentiation, the levels of some accessory subunits change dramatically, in particular those of the family of PCL family of proteins with DNA-binding domains. For example, the appearance of neural

progenitors occurs with downregulation of PCL2/Metal Response Element Binding Transcription Factor 2 (MTF2), the prevalent paralog in ESCs, and upregulation, instead, of Plant homeodomain finger protein 1 (PHF1) and PHF19. Likewise, the content in C17orf96/EPOP in PRC2 complexes diminishes. Perhaps more importantly, these stoichiometric modifications take place within a generalised reduction in the total level of PRC2 components (30–40 fold lower for core subunits). For PRC1 complexes, the levels of core subunits RING1A/RING1B and RYBP/YAF2 change little with differentiation, whereas those of PCGF or CBX subunits, as mentioned before, are affected significantly. Particularly intriguing is that most Polycomb E3 ligases in ESCs use PCGF6, unlike neural progenitors, enriched in BMI1–E3 ligases. Accordingly, in differentiated cells the contribution of PRC1.6-specific subunits, i.e., MGA, MAX, E2F6, to RING1B complexes drops while that of canonical subunits, PHC2, PHC3, CBX6, CBX8 increases.

## 3. Enzymatic and Non-Enzymatic Activities of Polycomb Complexes

The activity of the Polycomb system relies not only on the histone modifications induced by the catalytic modules of PRC1 and PRC2 (E3 ligase and KMT, respectively) but also on protein–protein interactions mediated by non-core subunits. These interactions, for example, are decisive for the formation of defined high-order and local chromatin structures. Both functions, catalytic and architectural, can be independent of each other. Additional enzymatic activities found in subsets of PRC complexes, such as CBX4-dependent protein sumoylation [60], or the histone H3K36 demethylase activity of KDM2B/FBXL10 [61] are not considered here.

### 3.1. PRC1 E3 Ubiquitin Ligase Module

The content in histone H2AK119Ub per cell is not known in a systematic manner, but work with cells lines indicates that it accounts for, approximately, 10% of total histone H2A [62]. This histone modification is almost totally dependent on PRC1 heterodimeric E3 ligases [63]. An exception is found in a subset of breast tumors and derived cell lines that downregulate RING1B and amplify the gene encoding Tripartite Motif containing 37 (TRIM37), an unrelated ubiquitin ligase [64]. Polycomb-dependent monoubiquitylation of histone H2A is conserved from plants and insects to vertebrates. The addition of ubiquitin (Ub), an 89 amino acid polypeptide to another protein is a multistep, orderly process that involves, successively, a Ub-activating component (E1), a carrier (E2) and a third element that provides substrate specificity (E3) [65]. PRC1 complexes contain the E3 ubiquitin ligases that target activated Ub bound to the E2 carrier to nucleosomal histone H2A. In vitro, the E2 ligase preferred by mammalian PRC1 complexes is UBCH5/UBE2D [19], although in vivo the critical E2 ligase is UbE2E1/UBCH6 [66]. Of relevance, the E2 ligase contacts heterodimeric E3 PRC1 ligases exclusively through the RING1A/B moiety, in sites away from the PCGF component [67]. In the chemical modification of histone H2A, the C-terminal glycine of Ub is linked to the amino moiety of K119 (K118 in *Drosophila*), and to equivalent lysines in variant histone H2AZ [68,69].

PRC1 recognition of the nucleosomal substrate relies on an interaction (arginine anchor) shared by many other chromatin modifiers [70]. RING1–PCGF heterodimers in PRC1 complexes form a saddled-shaped structure [67] that contribute a critical arginine (in the RING1 moiety) for a stabilised contact with the acidic patch formed by clustered glutamate residues from histones H2A and H2B [71]. Mutations in these histone residues result in dramatic decreases in the extent of PRC1-mediated monoubiquitylation [72]. Moreover, all six RING1–PCGF heterodimers fail to modify nucleosomal H2A in the presence of an acidic patch binding peptide [73]. Altogether, specific H2A monoubiquitylation results not only from the stereospecific PRC1 E3-nucleosome interaction but also from contacts between the E2 component bound to PRC1 and the histone C-term, that combined restrict the space susceptible to modification [72]. Interestingly, PRC1 E3 ligases found in non-canonical complexes are constitutively active, while those in canonical complexes are auto-inhibited, becoming active only after appropriate association with the nucleosome [73]. Divergent Ub-interacting interfaces in canonical and non-canonical PCGFs appear to determine the intrinsic activities of distinct PRC1 complexes [73].

Despite the structural characterisation of the complex between PRC1 catalytic modules and the nucleosome, the description of the modification process is still work under progress. For example, in vitro studies indicate that efficient monoubiquitylation requires auto-polyubiquitylation of RING1B [74]. Steady state levels of H2AK119Ub result from the antagonistic activities of PRC1-dependent ubiquitylation and that of a number of deubiquitinases that reverse the modification (mammalian BAP and Ubiquitin carboxyl-terminal hydrolase 16 (USP16) are two of them) in both global and targeted fashion. Another regulatory layer on the Polycomb-dependent modification of H2AK119 is achieved through post-translational modifications (specific phosphorylation) of the E2 carrier, or of RING1B itself, both of which have been shown to interfere with the ubiquitylation process [32,75]. This could explain the absence of H2AK119Ub-marked nucleosomes at sites occupied by PRC1 complexes [76].

### 3.2. Lysine Methyltransferase Module in PRC2

Mono-, di- and trimethylated H3K27 are modifications mediated uniquely by PRC2 subunits EZH2 and EZH1. The reactions are catalyzed by SET domains (named after three *Drosophila* proteins where the activity was first identified: Suppressor of variegation 3–9, Enhancer of zeste and Trithorax [77]), present in most KMTs. Unlike most other SET–KMTs, EZHs are catalytically active only in association with core subunits, EED, SUZ12 and RBBP4/7. PRC2 KMTs, thus, are obligate modular complex(es). A feature of KMTs is that cofactor methyl-donor S-adenosylmethionine and the peptide substrate bind distinct sites of the SET domain. However, unlike other SET–KMTs, in EZHs the two subdomains that make the catalytic region, the I-SET (inserted within SET, that binds substrate) and post-SET (immediately C-terminal to the SET fold, that binds methylation cofactor) are positioned in a non-functional configuration [78]. Its activation requires that EED and SUZ12 contacts, relayed through a SET activating loop in EZH, induce movements in these two halves priming them to position substrate and cofactor in a catalytically competent conformation. Thus, with the exception of the subset of non-canonical PRC1 E3 ligases, the catalytic activities of PRC complexes are regulated by their release of an auto-inhibited state.

The predominant product of PRC2 activity is the dimethylated form of H3K27 [79–81]. Quantitative proteomic estimation in ESCs identifies 50–60% of all histone H3 as K27me2, and 10–20% of the remaining H3 marked as K27me1 or K27me3 [82], thus making H3K27me2 one of the most pervasive histone modifications. The functionally significant Polycomb feature, the addition of the third methyl group, is a kinetically restricted modification [83], resulting in a localised distribution of the H3K27me3 mark. The tyrosine 641, in the EZH2 SET domain, facilitates the orientation of K27me0 and K27me1 to promote their methylation, while restricting further modification of K27me2 [83]. The normal preference of EZH2 for K27me0 and K27me1 substrates is altered in mutants whose SET domains modify extensively K27me2 leading to aberrant patterns as seen in lymphomas [83]. The kinetic barrier to the K27me3 state is overcome through a conformational alteration, triggered by EED binding H3K27me3, an allosteric change distinct from that releasing EZH2 from inhibition and that greatly stimulates its di- and trimethylase activity [84–86]. The proposed PRC2 feed forward activity, where a modified nucleosome stimulates the modification of the adjacent one, is supported by structural evidence showing reconstituted PRC2, with both reader (EED) and writer (EZH2) activities on the same complex, simultaneously contacting adjacent nucleosomes [87]. This permits H3K27me3 deposition over large genomic distances [82], the spreading of which is hampered in cells heterozygous for a H3K27M mutation [88]. Maintenance of PRC2-marked chromatin upon DNA replication it is thought to rely precisely on this ability to use parental nucleosomes as primers to propagate and restore the H3K27me3 state on new nucleosomes in the daughter cells [89]. Precisely, the different allosteric response associated to stimulated trimethylation of EZH1 and EZH2 explains the lower activity of EZH1-complexes [35,90,91]. Although their SET domains are well conserved, divergence in other regions of the paralogs make activating contacts with EED ineffective in EZH1. As a consequence, while

PRC2 complexes with either paralog catalyse mono- and dimethylation reactions, the trymethylated state is mostly due to EZH2-complexes [58,59].

EZH2 catalytic activity is not only regulated by interrelated interactions among core components but also can be stimulated, indirectly, through non-core subunits. For example, the presence of AEBP2 in reconstituted PRC2 complexes activates K327me3 deposition on nucleosomal templates containing monoubiquitylated H2AK119 [92]. Likewise, PCL paralogs or JARID2, through stabilised proximity to the substrate [82,91,93]. Moreover, the stimulatory effect of H3K27me3 recognition by EED can be achieved by EZH2-dependent methylation of lysine 116 in accessory subunit JARID2 [94], which adopts a structure assimilable to that of the methylated histone H3 tail [95].

H3K27me3 spreading, and the associated effects on transcription are maintained in check by chromatin modifiers that antagonise PRC2 catalytic activity. Most relevant is the methylation of histone H3K36, which results in an effective inhibitor of H3K27 trimethylation [96,97]. Thus, inactivation of Ash1, the product of a *Drosophila TrxG* gene [98,99] or of Nsd1-2 and Setd2 in mammalian cells [100–102] is accompanied by unconstrained extension of H3K27me3 domains. The effect is phenocopied in human cells bearing an unmethylatable H3K36M mutation [101]. Since PRC2 association to chromatin and its enzymatic activity are separable functions [58] it is likely that H3K27me3-free, PRC2-occupied genomic sites [76] result from its catalytic inactivation on chromatin.

### 3.3. Architectural, Non-Enzymatic Functions

Sites occupied by Polycomb proteins, sometimes separated by long genomic distances, can be physically proximate, as seen both in *Drosophila* and mammals. Regardless of the experimental approach, whether conventional fluorescent in situ hybridisation [103,104], super resolution imaging and refined in situ hybridisation [105–107] or molecular identification of genomic contacts in cross-linked chromatin [108–111], the observations point at the formation of nuclear structures dependent, at least in part, on the presence of a subset of PRC1 subunits. Some of these instances of genomic clustering, probably those of a larger size, correspond to the so-called Polycomb bodies, long known as cytologically observable foci enriched in PcG products [104,112,113].

Two types of protein structures, present in subunits of canonical PRC1 assemblies, appear as major determinants of these clustering events (Figure 2). One is the sterile alpha motif (SAM), a ≃70 amino acids module found in >300 other proteins [114], a structure with oligomerizing properties, located at the C-terminal region of Sex combs on midleg (SCM) homologs, SFMBTs, L(3)MBTs (except for L3MBTL2) and Polyhomeotic homologs. The second structure is determined by sequences enriched in positively charged amino acids that do not acquire any defined three-dimensional structure. Present in many proteins, these intrinsically disordered regions (IDRs) are found in mammalian CBX2/M33 or in *Drosophila* Posterior Sex Combs (PSC) [115,116]. IDRs participate in multivalency-driven interactions forming highly dynamic spatial compartments through liquid–liquid phase separation (reviewed in [117,118]).

PRC1 SAM-mediated contacts results in hundreds of domains, comprising from small, 20–140 kb genomic segments [107], to much larger regions [104–106,119]. Typically, sites organised in these structures are transcriptionally repressed. In *Drosophila*, H3K27me3-marked domains form highly compacted structures in which chromatin intermixing occurs at a high rate while, at the same time, remains isolated from other nuclear domains [105]. These repressed chromatin structures acquire a more relaxed conformation upon PH downregulation [105,107]. Similarly, the dynamic changes in the number and location of genomic contacts that normally take place during neural specification of ESCs are significantly altered by PHC1 inactivation [107]. Of note, clustering mediated by these contacts is affected only partially by inactivating mutations in the core subunits (RING1 proteins or EED) of Polycomb catalytic modules [107], underscoring their independence on histone modifications. The essential architectural role of PHC proteins is visualized by the partial disorganisation of PRC1 clusters in ESCs cells expressing a PHC1 variant whose SAM domain retains only one of the two interacting surfaces [106,120]. By excluding possible secondary effects associated with depletion

experiments, this PHC mutant, likely acting as a dominant negative variant, highlights the relevance of PHC-dependent interactions in the formation of high-order chromatin structures. For example, while similar disruptive events are also appreciated in cells lacking RING1 proteins [103,108] they might result indirectly from PHC destabilisation induced by the absence of RING1B [121,122]. Some of the PHC-dependent contacts form structures topologically similar to genomic loops in mammalian cells (Figure 2A), although their boundaries do not correlate with CCCTC-binding Factor (CTCF) sites usually found in them [107,123]. In *Drosophila*, however, where CTCF loops are not as prevalent as in mammals, looped structures identified within Polycomb-repressed compartments have PC-enriched boundaries [124]. Whether this underlies the disruptive effects that PC inactivation has on high-order chromatin structures in *Drosophila* [125,126] or whether these are primarily dependent on PH and triggered indirectly is not known.

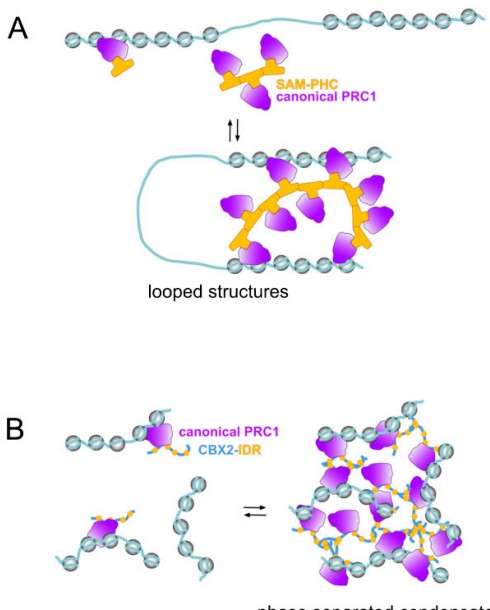

**Figure 2.** Polycomb and genomic compartmentalization functions. Characteristic of Polycomb activities is the ability to organize subsets of repressed loci in genomic structures where loci are clustered through singular protein–protein interactions between select canonical PRC1 subunits. (**A**) architectural functions mediated by SAM domain-containing subunits (Polyhomeotic paralogs, for example) underlie the formation of looped structures through contacts between genomic sites separated. The ability of every SAM domain to make contacts with two SAM domains is key to the stability of these structures. (**B**) an alternative source of clustered structures implies internally disorganized regions (IDRs) as those in CBX2/M33 in mammals, or PSC in *Drosophila*. Promiscuous contacts among IDRs, after achieving very high local concentrations gives rises to phase-separated condensates where the "trapped" genomic sites are physically separated from other nuclear components. Although transcriptional repression correlates with Polycomb condensates, these structures are not gene inactivation-specific (see [127]).

PRC1 IDRs were identified in a search for protein motifs responsible for the correlation between in vitro compaction of nucleosomal arrays and the inhibition of chromatin remodeling and transcription on defined templates by reconstituted/purified PRC1 complexes [128,129]. IDRs identified in the best compacting PcG proteins (determined by the significant contraction of internucleosomal distances in electron microscopy images) are *Drosophila* PSC and mammalian CBX2/M33 [115,116]. In particular, residues within the IDR in CBX2/M33 are critical for the formation of liquid phase separated condensates in vitro and in vivo (Figure 2B) suggesting a mechanism for the formation of Polycomb bodies [130,131]. Phenotypes associated to mutations in IDR-encoding segments in *Drosophila Psc* or mouse *Cbx2/M33* establish their functional relevance in vivo [132,133]. For example, a mouse strain engineered to

express a CBX2/M33 variant in which its charge was decreased by turning a number of lysines and arginines into alanines results in homeotic transformations of the axial skeleton, resembling those seen in other Polycomb mutants [133]. In this mouse line, no alterations in PRC1 occupancy or deposition of H2AK119Ub and H3K27me3 marks at PcG targets are observed [133], although accessibility, contacts or other measures of chromatin compaction were not assessed. In a subset of canonical PRC1 complexes SAM-containing and IDR-containing subunits coexist, which opens the possibility of a mutual influence between architectural functions mediated by these structures. Nucleosomal compacting activities have also been identified, in vitro, for PRC1.6 subunit L3MBTL2 or EZH1-containing PRC2 complexes [90,134] and whether they are related to the presence of possible IDRs or are biologically meaningful remains to be determined.

The above architectural functions of PRC1 align well with gene repression functions. However, Polycomb-dependent enhancer–promoter and promoter–promoter contacts are also involved in facilitating gene activation. For example, following EED depletion in ESCs, looped structures containing poised enhancers and promoters are severely affected, and although the enhancers do not become activated, the expression of genes (anterior neural) linked to them is defective in their differentiated progeny [111]. The requirement for gene activation, however, seems cell lineage specific as, in contrast to neural genes, mesoderm-specific genes are maintained repressed and subsequently activated as in wild type cells [111]. A related scenario is that of looping contacts between *Meis2* sites occupied by RING1B, at the promoter and downstream the polyA addition site, during mouse midbrain development. These interactions prevent a critical enhancer from reaching the promoter at early stages of development, while the release of RING1B from the promoter, at a later time, permits *Meis2* gene activation [135].

## 4. Polycomb Localisation on Chromatin

The range of residence times (on chromatin) for Polycomb subunits is similar to that of conventional transcription factors [136]. Associated in a highly dynamic manner, only a minor fraction of the pools of PcG products in the cell are bound to chromatin in a given time [137–139]. Chromatin maps of PRC1 and PRC2 subunits, or of the PcG-dependent histone modifications overlap at many but not all genomic sites, in cell context-dependent patterns [34,76,140,141]. Most mapped binding sites, in the widely studied ESC model, locate to promoters and intragenic regions [140,141]. However, unlike pluripotent cells [142,143], in differentiated and transformed cell lines PcG proteins occupy intergenic sites, often in regions corresponding to superenhancers (genomic regions with densely clustered enhancers) [34,144]. How Polycomb complexes "identify" genomic targets has been, and still is, a pressing question (see [145] for a review). A striking aspect is their association with both transcriptionally active and inactive genomic sites, even if in a different qualitative and quantitive fashion. As described below, until recently the main recruitment paradigms involved DNA-binding proteins (in *Drosophila*, mostly the products of non-PcG genes) and a hierarchical model in which PRC1 complexes were "hooked" onto PRC2-modified nucleosomes, through the recognition of H3K27me3 by CBX subunits. Currently, the localisation and activities of PRCs on chromatin is considered the outcome of a multitude of interactions involving DNA, histone tails, nucleosome and even RNA (Figure 3).

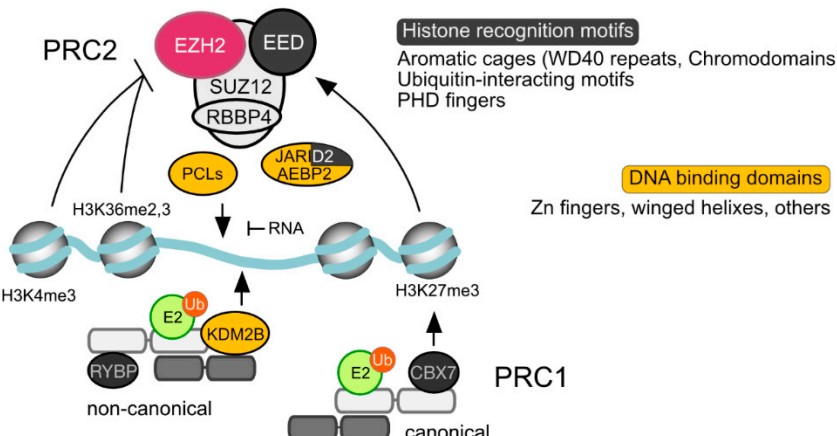

**Figure 3.** Polycomb subunits with motifs of relevance to chromatin recruiting. Schematized PRC1 and PRC2 complexes, showing subunits with proven roles in interactions with chromatin leading to extended residence times and/or nucleosome modifications. Lines indicate positive (arrowhead) or negative (blunted) regulation of catalytic activities/binding. Modified histone tails with clear roles in PRC2 activity/PRC1 binding, are indicated. Promiscuous interactions of RNA and PRC2 interfere with its ability to associate to naked DNA.

*4.1. DNA-Dependent Localisation*

Recruiting via DNA can occur either by direct DNA-mediated interactions, through a PcG subunit or, indirectly, through DNA-bound proteins that do not form part of Polycomb complexes. Direct recruiting to targets has been an elusive mechanism because during some time because *Drosophila* pleiohomeotic (Pho) was the only subunit known to encode a bona fide DNA-binding protein [142]. Pho, identified by its binding to the so-called Polycomb Response Elements (PREs), specific DNA regions able to confer repression to reporter genes in transgenic flies (reviewed in [146,147]), plays an important role in Polycomb recruiting [148]. In mammals, however, while the Pho homolog, Ying-Yang 1 (YY1), also binds DNA through the four shared C-terminal Zn fingers [142,143], its activity is little related to Polycomb recruitment [149,150].

Pho can be isolated as part of a complex—Pho-repressive complex (PhoRC)—in which MBT domain-containing Sfmbt is also present [151]. As PcG proteins with SAM-domains can interact with MBT domains, DNA-bound Sfmbt acts as an anchoring element recruiting PRC1 complexes to PREs [148]. PRC2 complexes, in turn, can associate to Sfmbt sites through interactions with SCM [152]. PhoRC subunits Pho and Sfmbt co-occupy most of the functionally identified PREs [153]. Interestingly, while many PREs localize to proximal promoter regions, a considerable proportion of those at promoter distal sites overlap with developmentally defined enhancers [153]. The stepwise recruitment to Pho sites is not conserved in mammals. The low affinity of YY1 for MBT-domain proteins [154] precludes a similar involvement in indirect recruitment as described in *Drosophila*. The closest PhoRC-related complex in mammals, PRC1.6, shares some of subunits, including MBT-domain protein L3mbtl2. However, instead of YY1, it uses as DNA-binding proteins MGA–MAX or E2F6-TFDp1 heterodimers [30,155–157]. The role of L3MBTL2, that together with PCGF6 play important roles in PRC1.6 targeting [30,155,156], is also different from that of Sfmbt because it lacks a SAM domain able to mediate subsequent contacts with PRC1 complexes.

Despite the important recruiting function of PhoRC, Pho binding by itself is insufficient, and additional PRE-binding proteins are required. Unlike Pho, these DNA-binding proteins (Spps, Dsp1, GAF, Combgap, Adf1, and others [158–160]) do not copurify with Polycomb complexes. Some of them, however, coimmunoprecipitate with PcG subunits [160,161], suggesting interactions that although not as strong as to withstand biochemical purification, can be functional at the densities and concentrations expected in nuclear environments. An indication of the oversimplification is that current models of

recruitment to PREs are oversimplified is evidenced by the PRC1-independent, Combgap-dependent localisation of PH [161]. The continuous reload of Polycomb components, needed to nucleate and spread PcG-modified chromatin surely uses a multiplicity of little known contacting pathways with PRE-bound proteins. In fact, effective recruiting can be attained even through the cooperation of PREs of low occupancy by DNA-binding proteins [162].

Direct recruiting of vertebrate PRC complexes through PcG DNA-binding proteins has come along rather slowly. For example, PRC1 recruitment through MGA–MAX and E2F6–TFDp1 heterodimers, mentioned before, has been verified only recently, despite their presence in PRC1.6 being known for a long time [163]. In other cases, unappreciated DNA-binding motifs have been identified in known PRC2 subunits [93,164]. Currently, most DNA-mediated Polycomb localisation involves direct binding through PcG subunits, although examples of indirect recruitment, through interactions with conventional transcription factors (for example, Runx1 in mammalian hematopoietic cell types [165]) are known. In agreement with the prevalent association of mammalian PRC subunits to GC-rich sequences with a high density of non-methylated CpG dinucleotides [149,166,167], specific protein motifs with these DNA-binding properties have been identified in PcG subunits. These motifs are of two classes: a CXXC-type of zinc finger present in non canonical PRC1.1 subunit KDM2B/FBXL10, similar to those in other chromatin modifiers [168,169], and a winged helix motif present in Polycomb-like (PCL) homologs in PRC2.1 complexes [93,164,170]. PRC2 subunits JARID2 and AEBP also contain DNA-binding motifs.

In vertebrates, these unmethylated CpGs concentrate on singular genomic regions known as CpG islands (CGIs), in the proximity of a large proportion of promoters [171,172]. A subset of these CGIs, usually occupied by PcG products, are marked with both H3K4me3 and H3K27me3. These regions, of low transcriptional activity are known as bivalent domains, due to the correlation of these histone modifications with transcriptionally active and inactive states. Modified by TrxG-encoded histone H3K4 HMTase MLL2, as part of one of the COMPASS (Complex Proteins Associated with Set1) complexes [173,174] and by PRC2, bivalent domains, first identified in ESCs [175], form and resolve dynamically during differentiation in a tissue-dependent manner [176,177].

The affinity of PcG DNA-binding motifs for CGI sites is but one of the multi-contact steps involved in recruitment. For example, PRC1 localisation in ESCs after KDM2B/FBXL10 downregulation is affected only partially, as shown by a limited decrease of RING1B occupancy [141,178–180]. The sites most sensitive are those with the lowest RING1B density, consistent with the involvement of additional localisation signals. KDM2B binding to chromatin is strictly dependent on its CXXC domain, as it is the recruiting of its direct interactor PCGF1 [141,178–180]. CGIs occupied by a high density of Polycomb products are enriched in KDM2B and not in its closely related paralog KDM2A, while the remaining CGIs are equally enriched in either of the paralogs [141]. Whether *Drosophila* KDM2B homolog, KDM—a component of dRAF complex [181]—plays any role in PRC1 recruiting has not been determined.

The second DNA-binding domain in Polycomb proteins that recognise GC-rich sequences is formed by a region contiguous to the C-terminal end of the second Plant Homeodomain (PHD) domain of PCL subunits, which folds as a winged helix motif [93,164]. PRC2 complexes reconstituted in the presence of PCL subunits show high affinity for double stranded DNA, have an extended residence time on nucleosomal templates and show increased activity in H3K27me3 deposition [93]. All these functions align with the critical role in vivo of PCL2/MTF2, the prevalent PCL paralog in ESCs [82,170]. Indeed, following acute depletion of MTF2, a substantial decrease in H3K27me3 density at sites normally occupied by EZH2 is observed, together with reduced binding of EZH2 [170]. By prolonging proximity with the substrate, de novo trimethylation of H3K27 could be achieved in conditions in which PRC2 cannot be allosterically activated by the absence of pre-existing H3K27me3. Thus, in *Drosophila* mutants carrying a null *Pcl* allele, the loss of H3K27me3 occurs with little alteration in the levels of H3K27me1,2 [182]. The affinity of mammalian PCL proteins for DNA, just as that of KDM2B/FBXL10, is sensitive to methylation [164,170]. However, unlike the lack of selectivity in

CGI binding by KDM2B/FBXL10, MTF2 associated preferentially to unmodified DNA sequences with a reduced DNA helix twist [170], is somehow characteristic of PcG targets. It has been suggested that methyl-free DNA recognition by these PcG subunits may be an evolutive adaptation in parallel with the selection of CGI sequences as genomic templates for gene control in vertebrates [171,172]. In contrast, *Drosophila* PCL binds DNA with no preference for non-modified sequences [93], perhaps related to the absence of CGI-related sequences in flies. An idealized rendering of de novo recruiting to chromatin, through DNA-binding motifs in PRC1 and PRC2, as well as their stabilization through mutual cooperation is shown in Figure 4.

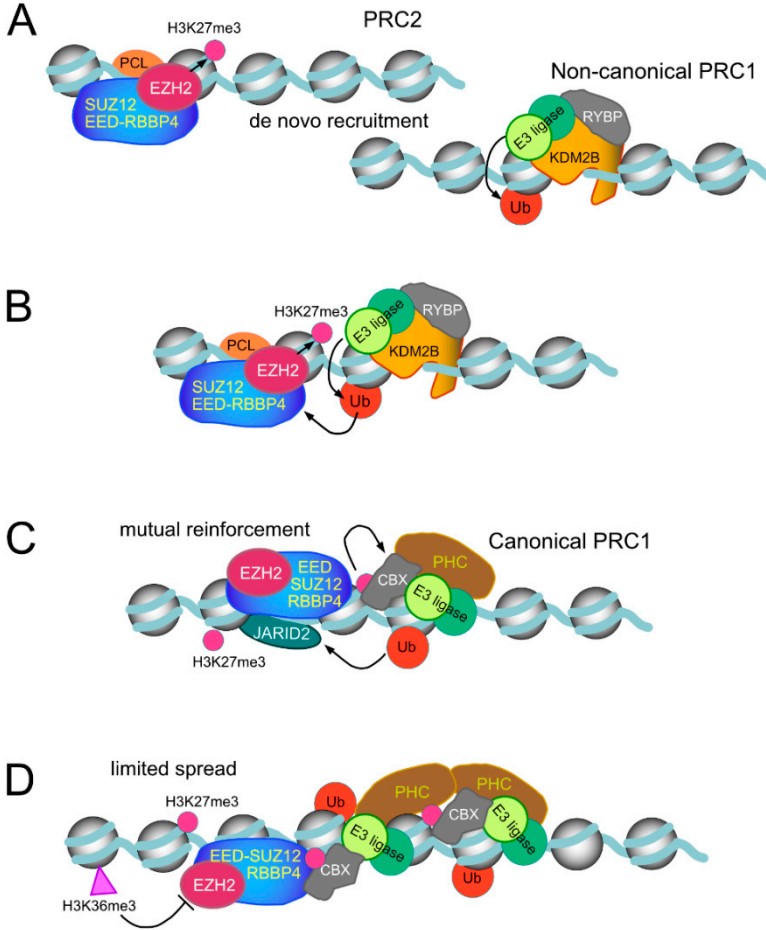

**Figure 4.** Idealized models of de novo recruiting of Polycomb and maintenance/restriction of their activities on chromatin. (**A**) Non-interfered affinity for DNA, winged helixes (PRC2 subunits, PCL) or CXXC-type zinc finger (non-canonical PRC1 subunit KDM2B) achieve residence times that enable the modification of nearby nucleosomes. Prevalent dimethylated H3K27 is the most probable substrate PRC2 complexes would find. (**B**,**C**) Display of possible stepwise reciprocal stimulatory activities of H2AK119Ub on PRC2 catalytic activity and of H3K27me3 promoting recruitment of canonical PRC1 complexes through chromodomains in CBX subunits. In addition to these collaborative interactions, recruiting and modifications by PRC1 and PRC2 can occur independently of each other. (**D**) Unchecked spread of H3K27me3 modification, based on the feed forward activity of PRC2, is restrained by the inhibitory effect of H3K36me2,3-modified nucleosomes. PRC1-dependent modifications undergo a quick turnover under the action of histone deubiquitinases.

The involvement of JARID2 and AEBP2 in PRC2 targeting is mediated only in part by their DNA-binding activities. JARID2 localizes to GC-enriched genomic sites [183] and, together with PCL2/MTF2, is essential for de novo nucleation of PRC2 sites [82] and for PCL2/MTF2 binding at

a subset of sites [170]. JARID2 shows a general affinity for DNA, possibly mediated by an AT-rich interaction domain (ARID) and a carboxyl terminal C5HC2 zinc finger [184]. In vivo, however, JARID2 and core PRC2 occupancy of Polycomb targets depend only on the ARID domain [185]. In the case of AEBP2, its DNA-binding activity depends on its three C2H2 zinc fingers [186]. Surprisingly, AEBP2's affinity for DNA increases if DNA is methylated [187,188] providing an explanation for the presence of H3K27me3 marks on sites with methylated CpGs [189,190]. Mutant ESCs expressing a truncated, C2H2 zinc finger-less AEBP2 show little alterations in PRC2 localisation and H3K27me3 marks [45], but in these cells most Polycomb targets are not methylated. In contrast, in differentiated cells, H3K27me3 density at DNA methylated promoters (*Hox* genes, for instance) decreases upon inactivation of Dnmt1 [191], the major maintenance DNA methyltransferase.

It is also worth noting a DNA-binding activity in SUZ12, defined by structures in its N-terminal moiety not folded in a recognisable motif that determines occupancy of CGIs in ESCs, independently of EED or EZH2 [192].

## 4.2. Interactions with Modified Histone Tails and Nucleosomes

Some PcG subunits, in both PRC1 and PRC2, contain protein motifs for recognition of modified histone tails, including the products of their own enzymatic modules (Figure 3). This property has been suggested to contribute some specificity to interactions with chromatin, considering the abundance of nucleosomes in the nuclear environment [193]. Thus, aromatic cages in CBX and EED subunits recognize nucleosomes with H3K27me3 [84,194,195] whereas those containing H2Ak119Ub can be recognized specialized domains in RYBP or YAF2 and JARID2 [196]. For PcG-independent modifications, Tudor-motifs in PCL subunits can act as binding methylated H3K36 [197–199] while interfaces in RBBP4/RBBP7 are sensitive to the methylation status of H3K4 [96,200]. Finally, uncharacterized determinants on the nucleosomal particle, rather than their histone tails, have been shown relevant for the JARID2-mediated PRC2 recruiting activity independent of its DNA-binding potential [91].

### 4.2.1. Polycomb Subunits that Recognize Histone H3K27me3 Tail

The widely accepted notion of H3K27me3-dependent interactions as a major PcG recruiting mechanism originated from work with *Drosophila* PC [11]. As other methylysine binding proteins, the chromobox domain of PC accommodates the methylated histone tail in an aromatic cage [201]. Mammalian CBX proteins, in contrast, do not have a preformed cage but it is induced upon binding to the modified histone tail [202]. The fact that chromobox domains are conserved throughout PC homologs and that the localisation of CBX proteins correlates with the H3K27me3 mark facilitated the extension of the recruiting paradigm into vertebrate complexes containing CBX subunits. Indeed, genomic relocation of PRC1 subunits, mirroring the extended deposition of H3K27me3 in cells that express K36M mutations at histone H3 have been observed [101]. However, the affinity of mammalian CBXs for H3K27me3 varies among paralogs (CBX7 and CBX6 being at the higher and lower ends of the range, respectively) and overall is well below that of *Drosophila* PC. Some mammalian CBX proteins even show no preference over H3K9me3 tails [203]. Given the similarity of mammalian CBX domains, including the amino acids that define the aromatic cage [194,204], the different abilities to interact with H3K27me3 may be dictated by the accessibility of the cage within the complex [202], perhaps better exposed in CBX7-PRC1 than in CBX6-PRC1, for example. Live imaging studies in mouse ESCs confirm that H3K27me3 participates in chromatin association of CBX7 and CBX8, but little in that of CBX2, CBX4 and CBX6 [138]. Moreover, when chromobox recognition of H3K27me3 facilitates localisation to chromatin, it is in cooperation with DNA contacts mediated by the adjacent AT-hook-like motif that acts as a functional DNA-binding module [138]. In summary, PRC1 localization through CBX-H3K27me3 interactions, at least in mammals, is probably more restricted than the wide acceptance of the mechanisms would imply. Moreover, recent work characterising interactions between CBX7 chromobox and H3K39 KMTases such as SETDB1 or EHMT1/2, points at indirect, H3K27me3-unrelated recruiting mechanisms [49].

The interaction of PRC2 with H3K27me3 uses an entirely different aromatic cage, defined by residues of the blade-folded WD-repeats of EED [84,195]. More than as a recruiting device, it seems this interaction has evolved to act as mechanism for the catalytic activation of PRC2. Indeed, EED cage mutants are less efficient in deposition of H3K27me3 in flies and mammals [82,84]. Imaging studies, on the other hand, show that disruption of the H3K27me3-EED interaction (by the small molecule A-395 [205]) has little effect, at least in the short term, on PRC2 chromatin localisation [139]. Instead, it appears that de novo methylation of pre-existing H3K27me2 is carried out by EZH2 activated after EED-H3K27me3 interaction at sites defined by PRC2 DNA-binding subunits, with adjacent nucleosomes becoming modified through iterated catalytic stimulation following EED-H3K27me3 contacts [58]. This notion is structurally supported by the cryo-electronmicroscopy images of a PRC2-dinucleosome complex showing how H3K27me3-EED contacts a nucleosome while the SET domain of EZH2 lies in the proximity of the H3 tail on the adjacent nucleosome [87].

### 4.2.2. Polycomb Subunits that Recognise H2AK119Ub

Monoubiquitylated histone H2A is a bulky modification recognised by the ubiquitin-binding zinc finger of non-canonical PRC1 subunit RYBP [206,207] (K63-diubiquitin, present in H2A and H2AX during the response to DNA double strand breaks, is another [208]). Evidence of the in vivo impact that H2AK119-RYBP interaction has on the functionality of PRC1 complexes is still unclear. For example, decreased levels of H2AK119Ub in RYBP-depleted ESCs occur without alterations in chromatin localisation of PRC1 subunits [26]. In contrast, the deposition of H2AK119Ub and H3K27me3 that accompanies ectopic *Xist* expression in an ESC model of X-chromosome silencing, occurs at a lower rate when the cells lack RYBP and YAF2 [209], and this alteration cannot be rescued by a RYBP variant with zinc finger mutations that impair its binding to H2AK119Ub [206,208,209].

Nucleosomal H2AK119Ub can also be recognised by PRC2 subunit JARID2. A ubiquitin interacting motif in the N-term of JARID2 is instrumental in PRC2 recruiting and H3K27me3 modification in a number of settings, including the inactive X-chromosome [196]. Such a JARID2-H2AK119 interaction could stabilise contacts with nucleosomes, explaining PRC2 recruiting to sites marked by H2AK119Ub [141,196]. It is not clear, however, whether this association would stimulate EZH2, because H3K27me3 deposition on nucleosomal templates containing H2AK119Ub is activated if the reconstituted PRC2 complexes contain AEBP2 [92]. Altogether, whether recognition of monoubiquitylated histone H2AK119 plays a general role or one specialized at singular chromosomal sites (as in X-silencing) remains to be determined.

### 4.2.3. Subunits that Sense the Methylated Status of Histones H3K4 and H3K36

Besides EED, PRC2 core complexes contain a second WD40 protein, RBBP4 (or paralog RBBP7), also present in other chromatin regulators, which binds the most terminal region of histone H3 [96,200]. This interaction, is compatible with SUZ12 association as it occurs in a PRC2 complex, but its affinity decreases when H3K4 is methylated [96,200]. Interestingly, deposition of H3K27me3 by reconstituted PRC2 is inhibited on symmetrically H3K4 methylated nucleosomes [210], by a reduced catalytic turnover, allosterically induced through SUZ12, rather than by an impairment innucleosome binding [96].

Mammalian DNA-binding PCL subunits in PRC2 complexes contain a Tudor motif that binds methylated K36 in histone H3 tail [197–199]. This modification, as H3K4me3, inhibits H3K27 trimethylation of symmetrically H3K36me2,3 modified nucleosomes. The effect, again dependent on SUZ12 [96], occurs with PRC2 complexes that lack PCL subunits [96,97,210] and, therefore, cannot be related to Tudor-H3K36me interactions. Such uncoupling of H3K36me3 recognition by PCL proteins is further substantiated by the inability of *Drosophila* PCL to recognise methylated histone peptides [198]. Regardless of an involvement of Tudor motifs, the antagonism between the methylation states of H3K27 and H3K36 is confirmed by the expansion of H3K27me3 domains that accompany decreased

H3K36 methylation, whether by inactivation of *Drosophila* Trx-G gene *Ash1* and murine *Nsd1* [102,211], which encode H3K36 KMTases, or by the presence of unmethylatable H3K36I or H3K36M [101].

Altogether, the interactions of Polycomb complexes with modified histone tails may influence residence times, in a recruiting perspective, but they appear to have evolved rather towards the modulation of their catalytic activities.

### 4.3. RNA Interactions in Polycomb Localisation

Another potential pathway for localization of Polycomb activity to defined genomic sites is through the recognition of chromatin-bound RNA. This notion, based initially on correlations between the presence of long non-coding RNAs (*lncRNAs*) *Xist* on the inactive X chromosome (Xi) or *HOTAIR* on the *HOXC* cluster and the recruiting of both PRC1 and PRC2 [63,212–215] is consistent with some biochemical evidence showing RNA-binding activity for subsets of PRC1/PRC2 subunits [216]. Subsequent work, however, indicates that direct association with RNAs is not as general a Polycomb recruiting mechanism as was once considered.

Now it is accepted that PRC2 complexes bind distinct RNA structures with variable affinities, in what it has been defined as promiscuous RNA binding [217]. Most of this activity is contributed by EZH2, through interspersed, non-canonical RNA-binding patches that recognize G-tracts-rich single stranded RNAs [218]. The interaction of highest affinity is one with four-stranded structures, G-quadruplex, resulting from a singular kind of pairing of repeated Gs [188]. For example, a *lncRNA* transcribed from telomeric repeats, *TERRA*, folds in a parallel G4 structure that is bound by a holocomplex made of EZH2, EED, SUZ12, RBBP4 and AEBP2 [188], and is essential for the characteristic modification of telomeric histones, including deposition of H3K27me3 [219]. However, although a role in direct PRC2 recruiting to chromatin cannot be discarded, the best understood function associated with RNA binding is the competition for PRC2 binding to DNA, thus preventing H3K27 modification at actively transcribed loci [188,220,221]. As for the *Xist*-related recruiting of PRC complexes to the Xi, now it is though to be indirect, mediated by the interaction of non-canonical PRC1 subunit PCGF3 with RNA-binding protein hnRPK, in turn bound to *Xist* [222]. Thus, PRC2 recruiting and H3K27 modification during silencing of the inactive X chromosome is proposed to occur, secondarily, after monoubiquitylation of H2AK119 by PRC1.3 [209]. *HOTAIR*-mediated recruiting, on the other hand, is rather controversial because the increase in H3K27me3 deposition associated with forced *HOTAIR* recruiting is independent of PRC2 [223] and also because of discrepancies in phenotype interpretation of *Hotair* mutant mice (discussed in [224]).

Concerning PRC1 subunits, and in line with previous studies reporting binding of certain chromodomains to RNA [225], all murine CBX proteins, except CBX2, are shown to retain RNA in band-shift assays [203]. Chromodomain-mediated RNA binding, however, is independent of amino acids that conform the aromatic cage that accommodates the methylated tail of H3K27 [226]. Functionally, a role for *ANRIL* lncRNA, transcribed from the *INK4* locus in orientation antisense to that of mRNAs encoding tumor suppressors (Ink4a/p14 (p16 in mice), Ink4aArf/p19 and Ink4b/p15) has been shown to participate in PRC1 recruiting to this well known Polycomb target [226]. Another example of RNA-PRC1 interaction, rather related with subnuclear localization than with localization to a given genomic site, involves subunit CBX4. In this case, depending on its methylation status, CBX4 binds either lncRNA *TUG1* or lncRNA *MALAT1/NEAT2*, resulting in CBX4 localization to Polycomb bodies or to interchromatin granules, respectively [227]. Recent RNA immunoprecipitation studies identify CBX7 binding to a number of RNA motifs in 3'UTR of mRNAs [228], although do not address an implication in PcG function.

### 4.4. Polycomb Eviction from Chromatin

Polycomb localisation on chromatin has been approached mostly by asking how the complexes interact effectively with their components and remain at sites long enough to induce histone modifications or other structural changes. Less studied is the opposite scenario, that of pathways

promoting release from chromatin. One obvious scenario is through post-translational modifications, of which phosphorylation-dependent release of PCGF4/BMI1 from chromatin in human HeLa cells is among the first examples [229] (or the opposite effect, promoting association, seen with PCGF2/MEL18 in murine ESCs [230]). A systematic investigation on the regulation of PRC association with chromatin, however, is still pending.

An alternative pathway involves active eviction of complexes from chromatin. One such mechanism, identified in the reversion of PRC1-modified chromatin states involves the H2AUb1-binding protein Zuotin-related factor 1 (ZRF1). Switching of silent, PRC1-occupied sites programmed to become active during retinoic-acid triggered differentiation of ESCs is accompanied by RING1B release in a ZRF1-dependent process [231]. Not a fully understood mechanistically, chromatin-recruited ZRF1 (through recognition of H2AK119Ub-nucleosomes) associates with CDK8 [232], a component of the Mediator, one of the highly complicated machines to be assembled to promote transcription (see [233]). This ability to displace RING1B is also observed, in a different context, linked to the monoubiquitylation of H2A at sites of DNA damage undergoing nucleotide excision repair [234]. The overall impact this type of chromatin eviction on Polycomb function, however, remains to be determined.

Another eviction mechanism involves the BAF (Brg/Brahma-associated factors) complex, a member of the group of chromatin-remodeling complexes whose DNA helicase belongs to the SNF2 subset of ATP-binding helicases of the DEAD/H family [235]. A measure of the physiological relevance of the interactions between BAF and PRC complexes is that four TrxG genes, at least, encode BAF subunits [3], including the DNA helicase BRG1/SMARCA4 (its *Drosophila* homolog BRG). It appears that, in contrast to other remodelers, the major substrate of BAF complexes are not nucleosomes. Instead, they participate in PRC1 and PRC2 eviction from chromatin. Oncogenic mutations in BAF components correlate with defective (SNF5/BAF47 inactivation, for example) or enhanced (expression of the product of a *SS18–SSX* translocation, leading to deregulated targeting) ability to evict Polycomb complexes from chromatin [236,237]. Thus, downregulation of BRG1/SMARCA4 in ESCs leads to extended occupancies of RING1B and SUZ12 subunits, including new, not previously bound sites [238]. Artificial recruitment of BAF complexes to select genomic sites suffices to displace PRC1 and PRC2 complexes in a quick (2–5 min), replication or transcription-independent manner [239]. Such an eviction requires of an intact ATPase activity, decisive at least in the stabilization of contacts between BAF core subunits and PRC1 components RING1B and RYBP [238]. Forced BAF1 recruiting (in mouse fibroblasts) leads to a swift release of RING1B and EZH2, with concurrent decreases in H2AK119Ub and H3K27me3 levels [239]. Major targets of BAF-related PcG eviction are enhancers and bivalent promoters [238,240].

Overall, the impact of components of Polycomb-eviction systems has been studied in a reduced number of situations. For example, neurogenic differentiation of ESCs is particularly affected upon ZRF1 depletion, whereas mesodermal or endodermal derivatives in teratomas derived from these cells are less affected [241]. ZRF1 is required for the formation of neural progenitors in vivo and in vitro, probably through sustained expression of Pax6 and other neurogenic transcription factors. In addition, their Wnt pathways-dependent self renewal, is maintained through Zfr1 promotion of the expression of Wnt ligands. Nevertheless, the contribution of PRC1 displacement to these functions has not been assessed, and additional mechanisms, such as an interaction with the retinoic acid receptor alpha, an important mediator of neurogenesis, has been described for Zrf1 [242]. For eviction activities dependent on chromatin remodelers, studies assessing Polycomb release associated to loss-of-function of BAF1 subunits or dominant negative mutants are mostly restricted to ESCs. Missense mutations, as many identified in cancers, correlate with transcriptional alterations at both Polycomb and non-Polycomb targets [238,243].

## 5. The Polycomb Machinery and Transcription Control

It is likely that cell context complexity has made a systematic characterization of Polycomb targets difficult, and a detailed description of known targets in diverse cell types is beyond the scope of this

review. Loci occupied or modified by PcG products are considered candidates for direct transcriptional regulation. Initial genome-wide studies of the DNA bound to PRC1 and PRC2 proteins, isolated by chromatin immunoprecipitation (ChIP) studies, identified sites in embryonic cell lines of flies and mammals corresponding to developmental genes activated during differentiation processes [244–246]. For example, Polycomb targets in ESCs belong into gene ontology categories such as development, organogenesis, morphogenesis and other similar [244]. Promoters of many of these targets are of the so-called bivalent category, a configuration particulary apt for permitting activation while maintaining a silent state in the absence of differentiating cues. Thus, even if variations in recruitment or other mechanistic processes are in place (CGIs versus PREs, for example) the regulatory logic served by PcG products is, in general, conserved.

A shared functionality of Polycomb targets in primitive cells, then, is the preservation of their self-renewal and developmental potential. This includes transcription factors with roles in cell commitment, and receptors and their ligands, of important signaling pathways (Wnt, Notch, FGF and others) [244–246]. Similar functions are also regulated by the Polycomb system in multipotent postnatal and adult cell types that regenerate tissues during life. Throughout a variety of mechanisms, mammalian PRC1 and PRC2 mutants undergo loss of stem cells. In part, proliferative arrest, triggered by derepression of negative regulators of cell proliferation encoded by the Ink4 locus, is one of the mechanisms; exhaustion of pools of stem cells, another. Some examples in neural [247,248], hematopoeitic [249], epidermal [250], intestinal [251] and other cell lineages. Likewise, skewed differentiation pathways are observed in some of these mutant mice models. For example, the conversion of hematopoietic T cells into B cells upon loss of suppression of the B-cell program in developing cells lacking PRC1 products [252], the reduced astrogenic fate of neural progenitors upon defective restriction of neurogenic competence of progenitors defective in PRC1 or PRC2 products [253] or the expansion of epidermal Merkel cells in the skin of Polycomb mutant mice [250]. These phenotypes correlate with deregulation of key transcription factors (Pax5, Neurogenin 1, and others important in epidermal development, respectively).

The study of better resolution and depth ChIP assays have subsequently expanded the range of Polycomb targets, both in flies and mammals, to include loci encoding products with metabolic, cytoskeletal and signaling functions [34,76]. These "atypical" targets are prominent in differentiated cell types, but subsets of them are also found in ESCs [254] and their contribution to any of the phenotypes associated with Polycomb mutations is not known.

How Polycomb transcriptionally regulates these targets can only be described, mechanistically, in rudimentary terms. In part, this is due to a still incomplete understanding of the mechanisms governing transcription [255] but also to the presence of Polycomb proteins in highly diverse transcriptional states, spanning from transcriptionally silent/poised loci to fully active sites.

Recent, unbiased analysis of Polycomb localisation unveils their presence on transcriptionally active promoters and enhancers besides their well known association with repressed/poised promoters. In fact, a large proportion of Polycomb targets identified in mammalian in vitro cell models correspond to transcriptionally active loci [29,32–34,76,256,257]. A similar scenario is observed in vivo. For example, in developing hematopoietic [57,75], spermatogenic [250] or skin [258] compartments. Localisation of PcG products to transcriptionally active loci is conserved in *Drosophila* [76,259–261].

On transcriptionally active sites, PcG products are said to perform non-canonical functions [57,76], in contrast to their better studied functions on silent targets. Typically, sets of canonical targets are enriched in members of gene ontology categories, such as developmental processes, whereas non-canonical, active targets typically belong in distinct Gene Ontology (GO) classes, such as cell proliferation/signaling and oncogenic pathways [34,76,144]. Different regulatory pathways for these two classes of targets is evident in *Drosophila*, where the canonical ones, decorated with H2AK118Ub and H3K27me3, are derepressed when genes encoding either PRC1 or PRC2 subunits are mutated, whereas non-canonic targets, lacking these marks, respond only to mutations in PRC1-encoding genes [76]. In addition to differing histone modifications, active and inactive targets diverge in

Polycomb occupancy densities. In general, the amount of PcG products bound to active sites is lower than those present at repressed targets. Another difference is that in cell types where Polycomb targets are canonically regulated, such as pluripotent mammalian cells or early fly embryos, PRC1 subunits concentrate on promoters, whereas in differentiated and transformed cell types, with a large number of non-canonically regulated targets, PRC1 locates preferentially to intergenic sequences [34,144]. Despite active mRNA synthesis, Polycomb proteins at these loci seem to play modulating negatively their transcriptional output, although in a subset of them, unexpectedly, they seem to act as positive regulators, promoting gene activity. A schematized view of Polycomb-bound active and silenced genomic landscapes is shown in Figure 5.

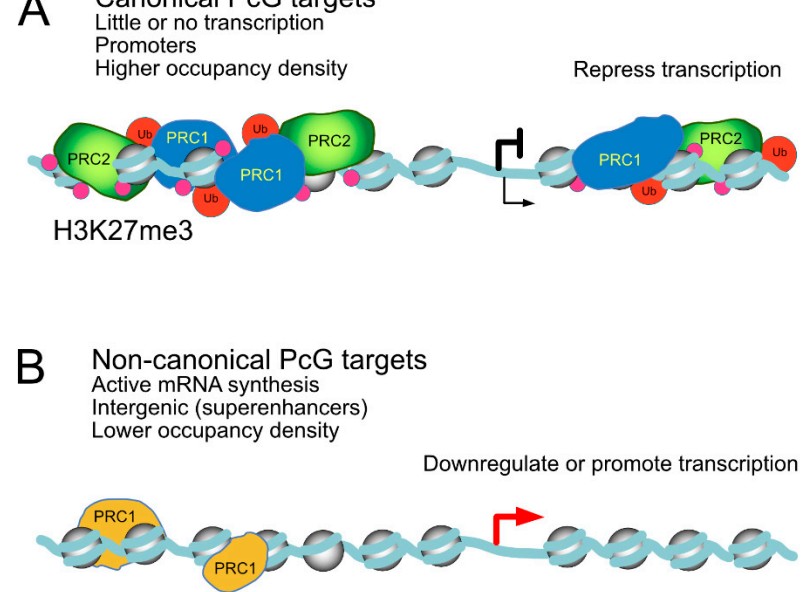

**Figure 5.** Comparison of Polycomb-bound loci with distinct transcriptional output. (**A**) canonical targets, where little or no mRNA synthesis occurs, show promoter-proximal regions highly enriched in both PRC1/PRC2 complexes and nucleosomes bearing Polycomb-dependent histone modifications, H3K27me3 in particular. (**B**) non-canonical targets, actively synthesizing mRNA, with regulatory regions away from promoters occupied by a reduced representation of Polycomb products, whose activity is somehow downregulated, leading to much lower levels of H3K27me3. Roles in downregulation/ upregulation at these targets are inferred from loss-of-function experiments.

*5.1. PcG-Dependent Transcriptional Repression*

Polycomb-dependent negative regulation of gene activity affects both transcriptionally active and inactive loci, corresponding most likely to two very different types of regulatory layouts. For historical reasons, however, the study of mechanisms that maintain repressed states has dominated investigations on Polycomb transcriptional functions. Perhaps under this influence, roles decreasing chromatin accessibility, supported by biochemical activities of PcG products, have appeared as the most important mechanism underlying Polycomb-mediated silencing. Indeed, although very scarce, the few studies that have addressed Polycomb function in global chromatin accessibility in vivo (assessed by the so-called assays for transposase accessible chromatin (ATAC)) show PRC-occupied promoters as less accessible structures than PRC-free promoters [261]. Another measure of chromatin compactness, the rate of nucleosomal density and its packaging degree, is higher for PRC-bound promoters. Unexpectedly, the correlation between gene activity and chromatin compaction is not regulated by Polycomb as much as it was expected. For exmple, chromatin accessibility is unaltered after genetic inactivation of both RING1A and RING1B or of EED, while nucleosomal structure while influenced by PRC1, is not influenced by PRC2 [261]. These results are consistent with the observation

that PRC1 localisation in cells with mutations in remodeler SMARCA4/BRG1, is augmented, possibly as a consequence of defective PRC1 eviction [238], but at sites whose accessibility is not altered [243]. This evidence refutes the widely accepted idea that Polycomb-mediated repression occurs through regulation of chromatin accessibility, at least as a general regulatory principle. It is not known whether these conclusions, withdrawn from genome-wide analysis, apply to PcG-dependent structures well correlated with gene inactivity, such as those generated through self-interactions (IDR-dependent, SAM domains) of canonical PRC1 subunits or other type of contacts [107,126]. It is certainly likely that the analytic tools used to determine chromatin accessibility and genomic contacts, ATAC, FISH, Hi-C and other methods, address molecular events not readily comparable.

The contribution of PRC1 to the nucleosomal configuration derives from the relaxed structure acquired by PcG targets in cells lacking both RING1A and RING1B, particularly at promoters of derepressed genes [261]. As increased nucleosomal occupancy and packing density occurs in normal cells upon inactivation of RNA polII, it is possible that PRC1 may act through prevention, somehow, of the transcription function. Some evidence along this line includes interference with the activity and/or assembly of the pre-initiation complex [262,263] or with the release of paused RNA polII [264]. Moreover, that transcriptional derepression in PRC2-mutant cells occurs despite an unaltered nucleosomal structure, possibly maintained by PRC1 action [261], further supports the uncoupling of chromatin compaction and transcription.

If chromatin accessibility and nucleosome packing are, at least in part, unrelated to Polycomb influence on transcription what about the histone modifications dependent on Polycomb complexes? It is generally accepted that they anticorrelate with transcriptional activity, and that nucleosomes at silent canonical, but not to active PcG targets are marked with H2AK119Ub and H3K27me3 [76]. The correlation would suggest a clear, direct or indirect, involvement of the modified histones in transcriptional activity. However, H2AK119Ub is a modification dispensable for repression at certain sites in flies and mammals [46,103,265,266]. Paradoxically, some of the regulatory machinery intended for gene activation contains histone H2AK119 deubiquitinases such as Myb Like, SWIRM And MPN Domains 1 (MYSM1) or BAP1 [267,268]. It is possible that targets modulated by these complexes undergo an "activating" step in relation to weakened H3K27me3 deposition subsequent to ubiquitin removal from H2AK119 [92,268,269], thus facilitating the likelihood of reversion of the silent transcriptional state. In fact, it is the trimethylation of histone H3K27 that best correlates with transcriptional silencing. The link was directly shown by the miss-expression of typical PcG targets (*Ubx*, *Abd-B*) and the homeotic phenotypes in mutant flies whose histone H3 has a K27 to R mutation, demonstrating that PRC2-repression uses this H3K27 modification [270]. How the deposition of H3K27me3 on nucleosomes relates to transcriptional inactivity is not known, but its repressive influence is underlined by a requirement for a demethylation step, attained through specific H3K27 histone demethylases (KDMTs) present in activating complexes, additionally endowed with histone acetyl transferases (HATs) [268,271,272] Since flies whose nucleosomes contain only H3K27R or H3K27A, variants that cannot be acetylated or methylated, derepress *Hox* genes in similar fashion to that of *E(z)* mutants [270,273], the role of H3K27 acetylation on gene activation is probably indirect, preventing repressive effects linked to the trimethylated state of H3K27.

Polycomb function(s) at transcriptionally active sites have not been studied in detail. PRC1 loss-of-function analysis in *Drosophila* (imaginal discs, tissue culture cells) shows upregulation in about a third of these PcG targets [76,259]. Similarly, RING1A and RING1B depletion in murine ESCs upregulates a subset of active targets [254,274], consisting of a repressive activity whose relation with that at silenced targets is not known. Another study, using differentiated murine myeloid cells, finds a correlation between genome-wide density of H3K27me3 marks and promoters transcribed at low levels [275]. Following depletion of canonical PRC1 subunits in *Drosophila* cells, changes in the phosphorylated states of RNA polII and decreased recruiting of Spt5 (a regulator acting in pausing and other transcriptional functions) are observed [259]. Alterations in transcription elongation rates and RNA stability add to the difficulties to interpret the regulatory scenario perturbed by

PRC1 inactivation [259]. Moreover, these targets may correspond with loci that show a high rate of transcriptional noise, flipping between alternate allelic forms: one, active and Polycomb-free, and another, Polycomb-bound and non-expressed [254,274]. The studies supporting this proposition, however, are very limited and further work is required.

## 5.2. Active Transcription Sustained by Polycomb

Contrasting with these repressive functions, both at silent and also active sites, a number of observations document a PcG requirement, mostly for PRC1 components, in gene activation (see Table 2; examples of indirect or non-PRC activities by Polycomb subunits, as in [276,277] are not indicated). In the absence of a systematic analysis, it is not possible to determine how general these functions might be. In some cases, activation correlates with co-occupancy of regulatory sites by RING1B and transcription factors critical for cell lineage identity, for example Spalt Like Transcription Factor 4 (SALL4) in spermatogonia [257] or Estrogen receptor alpha in breast tumor cells [144]. At least in tumor cells, some of the sites co-occupied by RING1B and particularly relevant to gene activation are superenhancers [144]. There, RING1B has been shown to adjust chromatin structure by both restricting and relaxing accessibility to superenhancers [144]. This relationship between RING1B and transcriptional activation appears rather unexpected despite documented examples of its presence in activating complexes containing TrxG subunits. For example, mammalian RING1B copurifies with MLL-1, 2-complexes in preparations containing HATs [278,279], or in complexes involved in histone acetylation and gene inactivation through an interaction with H3K27 KDMT Ubiquitously transcribed tetratricopeptide repeat, X chromosome (UTX), dependent on specific RING1B phosphorylation [32,280]. Beyond RING1B, in flies, subunits of protein complexes involved in gene activation, including members of the MYST family of HATs (MOZ/MORF), copurify with PC [50,260]. Interestingly, in human ESCs paralogs of scaffold subunits of these MOZ/MORF complexes, such as BRPF2/BRD1 colocalise with RING1B at a large number of bivalent sites [260]. Whether an involvement of PcG subunits in transcriptional activation is considered outside of the accepted characteristic Polycomb function—transcriptional repression—or not, these functions pose unanticipated difficulties to the genetic analysis and a challenging field for subsequent studies.

**Table 2.** PcG subunits involved in sustaining transcriptional activity.

| Subunit | Cell Type | Mechanism | Reference |
|---|---|---|---|
| CBX8 | Murine ESC-derived neural progenitor | not known | [281] |
| RING1B | Human melanoma lines | UTX, p300 recruitment | [280] |
| | Mouse spermatogonia, in vivo | Interaction activator (SALL4) | [257] |
| RING1A, RING1B | Mouse epithelial cells in vivo | not known | [250,282] |
| PCGF5 | Murine ESC-derived neural progenitor | not known | [283] |
| | Reporter constructs, established cell lines | p300 recruitment (AUTS2) | [32] |
| PCGF1 | Murine ESCs | not known | [284] |
| PCGF2/MEL18 | Murine ESC-derived cardiac-mesoderm precursors | not known | [33] |
| EZH1 | Murine myoblast line | RNApolII elongation | [56] |

## 6. Conclusions and Perspectives

Recent advances have enlarged the already substantial complexity of the "Polycomb system". It could possibly be said that, as a term, it is becoming too small to encompass the variety of biochemical entities, transcriptional mechanisms and targets regulated by their components. A realization that is becoming inescapable is that besides the familiar activities in gene repression, the system acts also on transcriptionally active loci, or so-called non-canonical targets. Likewise, the suspected correlation

between decreased chromatin accessibility and Polycomb-dependent repression of canonical targets is finding no robust experimental support. Thus, despite significant progress achieved in understanding interactions of relevance for recruiting to targets, mostly for PRC2 complexes, important questions remain open. Current insight derives from work with a reduced number of models and it is of a rather fragmented nature. For example, much of what is known about location of Polycomb subunits and associated histone modifications is correlative and lacking explanatory power. The highly dynamic deployment of Polycomb components and their association in various assemblies in a cell/lineage/developmental context-dependent manner is not matched with an understanding of the (suspected) functional diversity in the distinct settings. It is now clear that catalytic and non-catalytic functions can be unlinked, certainly for canonical PRC1 complexes. However, the actual role(s) of Polycomb-dependent histone modifications, even for those so well correlated with gene repression, such as H3K27me3, in any interaction with the transcriptional machinery are unknown. It is likely that progress in unraveling transcriptinal functions, particularly at a subset of active loci, will need simplified cell models and allele-specific analysis, so that the studies are not blurred by contributions of cell/allele diversity. Now that Polycomb is part of the current set of druggable biomolecules, addressing these questions may prove therapeutically beneficial.

**Author Contributions:** M.V. wrote the manuscript.

**Acknowledgments:** Work supported by a grant from the Ministry of Science, Innovation and Universities (SAF2016-80486-P).

**Conflicts of Interest:** The author declares no conflict of interest.

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
