# Peer review of "Polycomb Assemblies Multitask to Regulate Transcription"

_2075-4655, 2019_

Round 1

Reviewer 1 Report

The manuscript from Miguel Vidal addresses the role of Polycomb in transcriptional regulation. The author lists and names different Polycomb complexes based on their subunit composition and nicely addresses their enzymatic and non-enzymatic functions and the mechanisms leading to recruitment to targets. Overall, the review will be of a great value for the scientific community interested in gene regulation mediated by epigenetic mechanisms.

However, before publication, this reviewer suggests to expand the part related to Polycomb target genes. Some target genes are mentioned in line 186-190 or 799, but overall very little information is provided about what genes are actually regulated by Polycomb complexes at the transcriptional level. The author should highlight the main Polycomb target genes, discuss in what cell type or animal models they have been identified and how they are differentially regulated according to the subunit composition of the Polycomb complexes. The author should also discuss how mutations and alterations of Polycomb components or of genes which form part of the eviction mechanisms mentioned in section 4.4, affect regulation of target genes in development, cell-fate changes and cancer. This reviewer believes that a paragraph addressing the points mentioned above would be very useful and would be a good add to the manuscript.

Finally, it is very interesting what the author writes in line 132, namely that our information about PRC1 architecture is limited to a reduced number of cell types. It would be good to expand this point to both PRC1 and PRC2 architecture and function and highlight in what animal models and cell-types Polycomb complexes have been mainly studied so far and how we can use the data currently available in order to predict Polycomb targets and functions in other models and contexts of interest which would be worth studying more in detail in the near future.

Minor points:

Figure 1A: It would better to use Cbx 6,7,8 instead of 6-8

Table 1: There is a typo when non-canonical is spelled

Author Response

I appreciate the favorable impression on the review.

I do agree that no specific mention of Polycomb targets was missing. I decided this aiming for as much simplicity as possible (if at all it can be attempted within the large body of data available in the field). However, I realize, and therefore agree with the reviewer, that some general description of these targets, i.e. what Polycomb regulates, is in order, at least for readers with little or no familiarity with the system. I have added some paragraphs that convey common aspects of Polycomb targets, with examples of alterations associated to mutations in genes encoding Polycomb subunits. I'm afraid, that published studies dealing with null or other allelic variants generally do not include impacts on the biochemistry of the complexes, beyond the simple absence of a given subunit or the presence of a variant. Particular cases of allelic variants, aimed at testing a hypothesis had been included if in a dispersed manner throughout the text.

As for other aspects of Polycomb functionality, the impact of eviction systems has not been studied systematically. Nevertheless, I though it was important to include it because when thinking about occupancy/targeting, the prevalence of the binding side of the process makes it as if no active release participate/regulate the Polycomb landscape on chromatin. I have added a paragraph stating what, to my knowledge, can be confidently considered physiologically relevant on Polycomb activities, but it has to be considered that components of the eviction systems are not restricted to the Polycomb system.

Finally, new text is added to indicate most sources of data for PRC architecture, common for both PRC1 and PRC2, which, yes, are disappointingly scarce. That current data may have predictable power I find a bit optimistic. I rather think that the Polycomb system does not play instructive functions as, for example, transcription factors do. Apart from their activity on general functions such as cell proliferation or genomic stability (not mentioned in the manuscript because of the expressed focus on transcriptional control) I believe the system is more of a reactive nature.

Indicated typo (and additional ones) and text in figure 1, amended as suggested.

Reviewer 2 Report

In this manuscript, Dr. Vidal presents a complete and comprehensive view of the PcG complexes architectures and functions. The manuscript is very well written. The figures are very informative and complement the text very well. The only minor concern I have is that Table 2 appears to be slightly incomplete based on the discussed text. Also, the role of PCGF2 and PRC2 in activating gene expression in breast cancer should be also discussed (PMID: 25822021, PMID: 24055345, PMID: 28068325). Other than that, this is an outstanding review.

Author Response

Thanks for the positive assessment of the review.

I have some difficulty about the suggestions. In particular that within a general view of the Polycomb system in transcription, the activating roles of PCGF2 or EZH2 in breast cancer are not clearly linked to PRC1 or PRC2. Specifically, activating functions of EZH2 described in PMID: 28068325 and PMID: 24055345 are PRC2-independent; on the other hand,regulation of estrogen receptor a (ESR1) by PCGF2/MEL18 (PMID:25822021) occurs indirectly through the sumoylation of transcription factors required for ESR1 activation. These are important observations that I find difficult to use as examples of transcriptional activation by Polycomb, since it has been my intention to stick to functions of Polycomb subunits as part of conventional PRCs. In my opinion, including these, and other indirect, non-PRC functions would make for an unnecessarily complex manuscript. Instead, I am quoting two of these papers to illustrate this point.

Regarding the incompleteness of table 2, I'm not aware of a specific Polycomb subunit associated to transcriptional activation described in the text and not included in the table, but I may have missed it again.